# The Empirical Analysis of Environmental Regulation’s Spatial Spillover Effects on Green Technology Innovation in China

**DOI:** 10.3390/ijerph20021069

**Published:** 2023-01-07

**Authors:** Xinyu Wang, Yuanze Chai, Wensen Wu, Adnan Khurshid

**Affiliations:** 1School of Economics and Management, North China University of Technology, Beijing 100144, China; 2School of Economics and Management, Zhejiang Normal University, Jinhua 321004, China

**Keywords:** spatial econometric model, environmental regulation, green technology innovation, China

## Abstract

Green technology innovation is one of the driving forces of industrial structure upgrading. This innovation is thought to be related to environmental regulation. The study uses panel data for 30 Chinese provinces and cities from 2009 to 2020 and presents a comprehensive research-based explanation of how environmental regulations impact green innovation. This study employs the spatial Durbin model to analyze the spillover effect of the region. The results show that the total impact of environmental regulations is 0.223%, of which the direct effect is 0.099%. This impact includes the effects of both formal and informal environmental regulation. It indicates that ecological regulations significantly enhance green technology innovation. Furthermore, the spatial spillover effect is significantly positive at the 1% level with a coefficient of 0.124. Such spillover effects represent a learning effect of regional environmental regulation. Based on the results, the study suggests a few policy measures based on the detailed outcomes.

## 1. Introduction

The Chinese economy has entered a new normal, with new and old problems such as overcapacity, low effective utilization of resources, and challenging industrial upgrading. China was ranked the fourth worst country per global air quality in 2020, which is not optimistic for ecological and environmental protection. Carbon dioxide emissions in China are expected to peak by 2030, and the government is working toward carbon neutrality by 2060 [1]. The dual carbon targets are achievable with the support of technological innovation and the regulation of green and low-carbon [2]. Green has become the central economic and social development theme, changing the traditional innovation model to green innovation thinking. It can bring a win-win situation for the economy and environment of China as green technological innovation is of great significance to the sustainable development of human society, especially in an era of ecological pollution and increasingly depleted resources. Similarly, green innovation takes into account the functions of economic growth, environmental protection, and resource conservation [3]. Still, the negative externalities of the ecological environment seriously affect the industrial structure and other aspects. At the same time, the profit-seeking nature of enterprises makes it less likely for them to take the initiative to achieve green technological innovation. Therefore, there is an urgent need for external imposed or directed intervention, i.e., environmental regulation [4].

The Chinese government emphasizes environmental protection. It has implemented several policies to compel businesses to pursue green technological innovation. It includes the strictest environmental protection system proposed at the Fourth Plenary Session of the 19th Central Committee. The policy guidelines include administrative fines and various means to guide enterprises to implement multiple environmental protection measures, prompting them to green production and taking the road to green and healthy development [5]. Similarly, ecological behavior is gradually influencing the production and sale of green products as residents become more aware of environmental protection. The report of the 19th National Congress proposes “to build a market-oriented green technological innovation system” and “to strengthen the system support for green technological innovation through sound market mechanisms and flexible use of policy tools.” This shows that environmental regulation is an integral part of green technology innovation, but the effect of environmental regulation on green technology innovation is not yet known. Therefore, in the current context of high-quality economic development, it is essential to examine whether ecological regulation can promote green technological innovation and how effective it is. The outcomes will help investigate the rationality of environmental regulation and portray lessons for the coming years.

This study will analyze the impact of environmental regulation and green technology innovation spillover effect for 30 provinces and cities in China. The study will add to the empirical literature in the following ways. Regarding research intention, environmental regulations related to green technological innovation are divided into formal and informal categories. Furthermore, the impact of environmental regulation and green technological innovation varies by province. So, examining the spillover effect to test regions’ learning behavior is logical and can help in future decision-making. In terms of research content, the impact of environmental regulation on green technology innovation is limited to the regional perspective. Therefore, a spatial econometric model is used to investigate the spatial role of environmental regulations on green technology innovation from a broader viewpoint. The spatial Durbin model generates unbiased estimates of coefficients regardless of whether the correct data-generation procedure is a spatial lag or spatial error model. Furthermore, it does not limit the size of possible spatial spillover effects in advance, which is a positive aspect. At the same time, there is a tendency to simplify the use of matrices. This leads to a certain degree of bias in the analysis of spillover effects. This paper compares the regression results under the three matrices to demonstrate, however, the previous studies tend to do only 0–1 matrix analysis. This study incorporated spatial factors into the model. Furthermore, we consider economic and spatial weights in the neighborhood as they may support enhancing green technology innovation.

The remainder of the study is as follows. The next section explains the empirical literature, while Section 3 summarizes the theoretical literature. Section 4 has the material and methods used in this study; Section 5 explains empirical outcomes, and Section 6 concludes this study.

## 2. Literature Review

A growing body of literature examines the connection between environmental regulation and developing new green technologies. Environmental regulation and technological advancement research can be categorized into four broad groups.

The first view is that environmental regulation can significantly promote technological innovation. Before doing so, it is first necessary to clarify the policy’s objectives. Titanberg emphasized in the issue of sustainable development that even efficient markets may not be sustainable. Pollution control requires government intervention. On the issue of environmental policy, he argued that it is essential to focus on the issue of efficiency and cost-effectiveness. Porter [6] proposed the “Porter Hypothesis,” which argues that moderate environmental regulation significantly impacts technological innovation. This perspective gives a way of evaluating environmental regulation. That is, considering the role of environmental regulation in enhancing technology.

Many studies are trying to prove this point. Mbanyele and Wang [7] used a policy of air pollution reduction enacted in China under the 12th five-year plan in 2012. The outcomes using the difference-in-differences (DID) method show that environmental regulation is one of the potential channels to promote technological innovation. Song et al. and Frondel et al. [8] argued that the stricter the environmental regulation, the stronger the promotion of green technology innovation. Han and Chen [9] used meta-analysis to analyze the literature on environmental regulation and green innovation and found that environmental regulation in the Chinese context has a significant positive impact on green innovation. Qiao et al. [10] found that environmental regulation positively moderates technological innovation by applying fixed effects and system-GMM models. The findings portray that environmental regulations can effectively promote technological innovation.

Of course, proving that environmental regulation and technological innovation exist to influence is the first point. Many studies go further to discuss the heterogeneity of such effects. Li et al. [11] used panel data from 12 resource-based industries from 2003 to 2009 to construct an econometric model to examine the effect of environmental regulation on technological innovation in resource-based industries. The outcomes revealed that environmental regulation promotes technological innovation in resource-based industries. Song et al. [12] analyzed the spatial heterogeneity of the environment on firms’ technological innovation. It shows that formal environmental regulation can effectively promote firms’ technological innovation. Chen et al. classified 37 industries according to carbon emissions and R&D intensity [13] and found that environmental regulations positively affect R&D expenditure in both high and low-carbon and medium and low-R&D industries.

From the point of view of emission control science, the main thing that technological change brings is cost savings. In particular, the implementation of the same emission rates for different sources of pollution. This will result in firms minimizing their marginal pollution control costs through technological change. Due to the same emission rates, firms will independently choose their emission levels under a consistent level of marginal control costs. This is necessary information for regulatory agencies to develop policies. The regulator adjusts the emission rates based on the results of environmental controls. In some studies, informal environmental regulation has been found to have the same effect. Hille et al. [14] demonstrated that stricter ecological regulations promote innovation. Norberg et al. [15] discussed that environmental regulation facilitates technological innovation to some extent. Fang et al. [16] used a difference-in-difference approach to demonstrate that this supports a weak version of the Porter hypothesis in China. Hamamoto et al. [17] argued that the growth of investment development under strict regulation positively affects the total factor productivity growth rate. Taking the Chinese Environmental Protection Law as an example, high-intensity environmental regulation is more conducive to incentivizing listed companies to implement green innovation [18] but lacks spillover effects [19].

The second view is that environmental regulations are too harsh and hinder green technology innovation. They argue that environmental regulations increase firm costs and decrease firm productivity [20]. As it reduces green technology innovation, patent output falls, resulting in a drop in corporate earnings [21]. Therefore, neither the strong nor the weak Porter hypothesis is valid under this view [22]. Xie et al. examined data from 36 Organization for Economic Co-operation and Development countries from 2013–2018 [23]. They argued that environmental regulation volatility increases uncertainty and unpredictable risk for firms and investors, significantly impedes willingness to contribute to innovation activities, and leads to lower levels of innovation. He et al. [24] tested the Porter hypothesis using Driscoll–Kraay standard error estimates and various regression models and found that it does not hold in A-share listed companies. Yuan et al. [25] constructed an extended CDM model. They used panel data from 2003 to 2014 on Chinese manufacturing industries to test the impact of environmental regulations on industrial innovation and green development. The results did not support either the weak or the strong form of Porter’s theory.

The third point is that the relationship between environmental regulation and green technological innovation is not binary but rather diverse. Shao et al. [26] argued that ecological law significantly impacts various types of green technological innovation in enterprises. Still, there is a nonlinear relationship between the intensity of environmental regulation and some of the green technological innovations in enterprises (reducing the intensity of sulfur dioxide emissions, reducing the intensity of industrial wastewater emissions, and increasing the removal rate of industrial wastewater). Lv et al. [27] argued that there is regional heterogeneity in the impact of environmental regulation on green technological innovation; Xu Hui et al. [28] argued that environmental regulation has a positive linear relationship for low-carbon industries, an inverted U-shaped relationship for high-carbon industries, and a U-shaped relationship for the whole industry. Ouyang et al. [29], using a two-way fixed-effects panel data model to investigate marginal and heterogeneous effects empirically, demonstrated a U-shaped relationship between environmental regulation and innovation. Dong et al.’s [30] findings demonstrated that environmental regulations have a ripple effect, altering not only the development of green technologies in local regions but also those in adjacent ones. Cao [31] used a fixed-effects regression model to demonstrate a U-shaped relationship between environmental regulation’s strength and innovation level. From the perspective of industrial transfer, the relationship between environmental regulation and technological innovation is nonlinear. Zhang et al. [32] used a frontier method in the field of causal inference and showed that the effect of environmental regulation on the level of product innovation had a U-shaped relationship. Xu et al. [33], while studying 13 manufacturing sectors in China, found a nonlinear relationship between environmental regulation and green STI. Pan et al. [34] demonstrated an inverted U-shaped relationship between environmental regulation and green innovation based on panel data of industrial sectors in 30 Chinese provinces. Zhang et al. [35] applied data of industrial enterprises listed from 2012 to 2017 and showed a U-shaped relationship between environmental regulation and corporate innovation. Zhang et al. [36] demonstrated that public participation in environmental regulations positively affects cleaner production technology innovation. In contrast, market incentive and public participation rules have an inverted U-shaped relationship with end-of-pipe technology innovation. The effect of different types of environmental regulations on green technology innovation also varies [37,38], with market-incentivized environmental regulations stimulating green innovation but command-based environmental regulations inhibiting it [39].

The fourth view is that there is little interaction between the elements of environmental regulation and technological innovation. Li [40] tested the spatial effect and found that inter-provincial eco-friendly innovation exhibits aggregation attributes, and the government is pivotal to its advancement. Therefore, government environmental regulation based on green development will have an inverted “U” influence on environmentally friendly tech innovation locally. Chang et al. [41] tested Porter’s hypothesis and concluded that stricter environmental regulation could increase profits but would not stimulate firm innovation. Adam et al. [42] found little indication that the industry’s invention output was related to compliance costs and the Porter hypothesis by evaluating a representative sample of manufacturing environmental expenditures and innovation. Finally, by studying two important cases claimed by PH, Desrochers et al. [43] obtained that regulatory pressure is only a secondary factor and that environmental regulations are unlikely to promote industrial innovation.

The above analysis is the direct impact of environmental regulation and green technology innovation. Similarly, due to their close relationship, there is a Solow paradox between innovation and R&D investment [44]. Fernandes [45] found that investment in R&D does not necessarily lead to technological innovation and that there is an asynchrony between them.

Currently, spatial econometric models are also widely applied, especially the spatial Durbin model to solve real-life economic and social problems, including agriculture [46,47], industry [48,49], tourism [50,51], real estate [52,53], and so on. Moreover, the study of green development has been exceptionally well established, as Li and Wu [54] examined the spatial spillover effects of green technological innovation. However, at the same time, some areas need further discussion. This study examines the relationship between environmental regulation and green technological innovation, categorizing environmental regulation as formal and informal. Furthermore, each state and province experiences the effects of environmental law and green technology differently. As a result, looking at the spillover effect is crucial, as it can advise your decisions heading forward. There is a lack of national-level studies examining how environmental regulations have impeded the development of cutting-edge green technologies. As a result, ecological regulations’ effect on developing environmentally friendly technologies is studied using a spatial econometric model. This study utilized the spatial Durbin model to generate unbiased estimates of coefficients, irrespective of whether the correct data-generation procedure is a spatial lag or spatial error model. It also has the advantage of not putting a cap on the magnitude of any future spatial spillover effects.

Additionally, there is a trend toward reducing the complexity of matrix applications. This introduces some bias in the study of spillover effects. This paper compares regression results under three matrices to illustrate the spillover effects of environmental regulations. This research took into account spatial considerations. In addition, we consider the regional economic and spatial factors that may encourage further development of environmentally friendly technologies.

## 3. Theoretical Analysis

Environmental regulations play different roles in green technology innovation. Formal environmental regulation is a normative document with compulsory power issued by governmental institutions to improve the ecological environment and fulfill environmental protection duty [55]. On the other hand, informal environmental regulation represents the people’s will and is gradually increasing; green technology innovation needs long-term accumulation.

In 2006, China proposed a plan to build an innovative country. Since the 18th National Congress, the term “innovative country” has taken on a new meaning. It now includes building an ecological environment based on technological innovation, putting the market-driven development of green technological innovation ahead of the construction of an ecological environment, using a mix of economic and administrative tools to force or guide businesses toward technological innovation, and formalizing environmental protection. First, the government regulates companies by setting standards for pollutant emissions and charging specific fees for emissions to force businesses to develop green technologies. Environmental protection departments also require businesses to promote green manufacturing, improve green processes, and make green products, leading to the transformation of traditional industries like iron and steel, chemicals, and cement [56]. Second, the government compensates companies for losses they bear due to participation in environmental protection through financial subsidies or reducing sewage treatment costs. Third, enterprises respond to the government’s call to invest a lot of money in the first phase to buy environmental protection equipment. However, the upfront costs are not paid, which causes a “crowding out effect” [57]. When businesses put in place formal environmental rules and have less money, it could slow down the development of green technologies.

In recent years, informal environmental regulation has become increasingly popular among scholars, influenced by the market-oriented economy. The public’s ecological and environmental awareness has gradually increased, taking the initiative to supervise the government’s environmental protection enforcement scale and temperature. It exposes enterprises’ sewage flow, insufficient ecological protection, and approval procedures that cause enormous pressure on the government and enterprises. Informal environmental regulation influences green technology innovation mainly through the following ways [58]. First, residents negotiate directly with enterprises regarding environmental pollution and physical damage caused by their emissions, forcing them to take measures to reduce their polluting emissions. Second, through direct exposure to corporate pollution behavior, environmental organizations use Internet media and other means to protest ecological pollution, increasing negative corporate information and forcing companies to save themselves environmentally. Third, change the concept of consumption; the increase in demand for green consumption will attract the attention of enterprises. Therefore, they increase green investment by meeting the market demand and maximizing profits.

The exchange of economic activities is not limited to one corner, and there are spatial spillover effects. So, it is unreasonable to only look at direct effects when studying the relationship between environmental regulation and green technology innovation. On the one hand, according to the “pollution sanctuary hypothesis,” when the intensity of environmental regulations in the region is high, polluting enterprises will move to areas with weaker environmental regulations [59]. On the other hand, under the bottom-up competition effect, some regions usually relax environmental regulations, lower the ecological access threshold, and sacrifice the local environment to guarantee GDP growth. As a result, the polluting industries have less incentive to adopt green technology innovation. Insufficient motivation, thus not conducive to green technology innovation to enhance. On the other hand, innovation-led development runs through all aspects of the economy in the context of high-quality development. In different ways, the new development concept has guided the formation of environmental regulation policies worldwide. When a region has a strong capacity for green technology innovation, other areas will follow suit, creating a “learning effect.”

## 4. Materials and Methods

### 4.1. Fixed Effects Model

According to the theoretical analysis, we chose the panel test to examine the impact of environmental regulation on green technology innovation for the following reasons. First, the panel model can clearly illustrate the cross-sectional characteristics of the data while accounting for the serial attributes. Second, the individual effect model can control the problem of omitted variables that do not change over time. Therefore, the panel data of environmental regulations affecting green technology innovation are set as follows:(1)$$Ginn_{i,t}=\alpha +\beta Ier_{i,t}+\varphi Control_{i,t}+\nu_{i}+\varepsilon_{i,t}$$

In Equation (1), green technology innovation (*Ginn*) is the dependent variable expressed by the expenditure on new product development/energy consumption in high-tech industries. β reflects the impact of environmental regulation on green technology innovation, *i* and *t* denote region and time, respectively [60]. Similarly, $\nu_{i}$ is the individual effect and ε is the random disturbance term. Environmental regulation (Ier) is the core explanatory variable measured by the amount of investment completed in industrial pollution control/industrial value added. Table 1 is shown below.

For the calculation of environmental regulation intensity (*Ier*), there are generally several ideas. One of them uses the cost of operating pollution control facilities, or the per capita operating cost to measure. Or it takes the change in pollution emissions or the intensity of pollution emissions per unit of output. However, all these calculation methods run the risk of having too homogeneous indicators. Therefore, this study uses the share of pollution control investment in the increase of enterprises as a proxy variable. This allows us to measure the degree of individual compliance with formal environmental regulations [61].

The control variables (control) are foreign direct investment (FDI), fixed asset investment (Invest), fiscal expenditure (Fiscal), and urbanization (Urb). The FDI, Invest, and Fiscal, respectively, are determined by the percentages of GDP attributable to foreign direct investment, fixed asset investment, and budgetary share. At the same time, URB represents the urban population in total.

This paper refers Pargal’s research [62], which measures the intensity of informal environmental regulation in each US state using a combination of variables. The research found that people’s opinions about the environment improved with higher incomes, the transfer of government subsidies, exposure to the outside world, and higher population densities. For this reason, Foreign Direct Investment (FDI), Fiscal Regulation (Fiscal), Urban Regulation (Urb), and Environmental Investment (Invest) were selected as the intensity of informal regulation of the environment. The variable associations are shown in Figure 1.

### 4.2. Spatial Durbin Model

There is spatial spillover between regional technological innovation; if technological innovation in one region changes, it will impact technological innovation in neighboring regions. The same is true for green technology innovation, especially considering spillover effects. The existing literature mainly includes the spatial lag model (SLM), spatial error model (SEM), and spatial Durbin model (SDM) [63,64]. The spatial lag model is mainly used to explore the endogenous interaction effects among the explained variables. The spatial error model explores the interaction effects among the error terms, i.e., the magnitude and direction of the neighboring regions’ influence on the region’s variables. On the other hand, the spatial Durbin model accounts for the correlated error term and the interaction effects between the explanatory and explained variables, reflecting that one region’s green technology innovation influences its neighbors. Therefore, this paper constructs a spatial Durbin model based on the fixed-effects model to analyze environmental regulations’ direct and indirect effects on green technology innovation.

The spatial Durbin model of the effect of environmental regulation on green technology innovation is constructed as shown in Equation (2):(2)$$\ln Ginn_{it}=\alpha +\rho \sum_{j=1}^{n}\omega_{ij}Ginn_{it}+\gamma X_{it}+\theta \sum_{j=1}^{n}\omega_{ij}X_{it}+\nu_{i}+\varepsilon_{it}$$

In Equation (2), *X* is the independent variable containing core explanatory and control variables. ω is the spatial weight matrix and γ is the regression coefficient of explanatory variables, measuring the degree of influence of explanatory variables on the explained variables in the region (direct effect). θ is the spatial regression coefficient of explanatory variables, measuring the indirect impact of explanatory variables, i.e., spatial spillover effect. The spatial weight matrix is selected as a second-order inverse distance to measure. The distance between neighboring provinces’ capitals is calculated from each capital’s latitude and longitude information. Similarly, the reciprocal of the square of the distance is taken to measure the spatial correlation between the provinces. ρ is the spatial autoregressive coefficient and θ is the K* and the 1st order parameter vector.

## 5. Results

### 5.1. Descriptive Statistics of the Sample

This study examines the said relationships using the 30 Chinese province’s data from 2009 to 2020, obtained from the China Statistical Yearbook. The descriptive statistics of the variables are shown in Table 2. The outcomes in the table show that the variables vary widely among provinces in all years, with the minimum value of the green technology innovation level being only 0.236 and the maximum value being 671.758, a difference of more than 2800 times. The core explanatory variable, environmental regulation, also has a significant variation, with a difference of about 308 times between the minimum and the other minimum values. Taking the variables’ log values is a necessary first step before treating them for the large unit differences between them. This also eliminates heteroskedasticity and stabilizes the variables’ fluctuations.

### 5.2. Spatial Correlation Test

The spatial Durbin model is used because the variables are spatially correlated. Moran’s *I*, Geary’s C, and other standard spatial analysis methods are employed to determine whether variables are statistically correlated. The following equation is used for said purpose.
(3)$$I=\frac{n}{\sum_{i=1}^{n}\sum_{j=1}^{n}W_{ij}}\times \frac{\sum_{i=1}^{n}\sum_{j=1}^{n}W_{ij}\left(x_{i}-\bar{x}\right)\left(x_{j}-\bar{x}\right)}{\sum_{i=1}^{n}\left(x_{i}-x\right)}$$

In Equation (3), *i* and *j* denote the *i*th and *j*th regions, *W_ij_* is the spatial weight matrix, and the geographic weight matrix is selected in this paper. The value of *I* takes a range from −1 to 1. The larger the absolute value of *I*, the stronger the correlation. When *I* > 0, it indicates a positive correlation; when *I* < 0, it means a negative correlation.

In this paper, Moran’s *I* index is used for measuring the spatial relevance of green technology innovation from 2009 to 2020, and the obtained index is shown in Table 3. Table 3 shows that the green technology innovation Moran’s *I* value is between 0.249 and 0.99, and the *p*-values are all less than 0.01. Hence, it is suitable for analysis using a spatial econometric model.

Figure 2 depicts the scattered distribution of provincial green technology innovation based on Moran values. The positive autocorrelation of green technology innovation has existed for all years. The performance is characterized as follows: most provinces’ green technology innovation is located in the first quadrant or the third quadrant, and 12 provinces, such as Beijing, Tianjin, Shanghai, and Zhejiang, belonged to the first quadrant in 2020. In comparison, 14 regions, such as Hebei, Shanxi, Inner Mongolia, and Liaoning, are distributed around provinces with a high (low) level of green technology innovation, which belongs to the high-high combination and the low-low combination. This phenomenon implies that green innovation is often not in the form of clusters. It is more dependent on the influence of individuals. In this paper, we refer to this phenomenon as centralized green innovation. As for the reasons for the emergence of this phenomenon, we attribute it to policy. In particular, China has adopted the efficiency-first policy since its reform and opening up. This policy results in the priority development of some of the cities. They tend to be able to concentrate large amounts of resources to achieve industrial upgrading. In addition, the urbanization of a large population makes these cities invest more resources in the environment.

### 5.3. Model Selection

The most popular spatial econometric models, in addition to the spatial Durbin model, are the spatial autoregressive (SAR) model and the spatial error model (SEM). The three spatial econometric models are estimated, and the regression results are summarized in Table 4 based on Equation (2) to select the most appropriate model. The outcomes from the three models show that environmental regulation significantly positively affects green technology innovation.

The presence or absence of spatial interactions within the SDM model is an additional factor when deciding if the model is appropriate for this investigation. The results of the LR test are used to determine which model was suitable. Table 5 shows that all tests rejected the null hypothesis at the 1% significance level, indicating that the SDM did not degenerate into the SEM or SAR models. Therefore, the spatial Durbin model is used as the optimal model choice.

### 5.4. Spatial Spillover Effect

Table 6 shows the estimated results of the spatial Durbin model. Still, these results do not adequately depict environmental regulations’ direct, spatial spillover or total effects on green technology innovation. Therefore, Table 6 displays the estimated outcomes for the direct impact, spatial spillover effect, and total effect.

Table 7 shows that environmental regulations’ direct, indirect, and total effects on green technology innovation are significant, with coefficients of 0.099, 0.124, and 0.223, respectively. So, if environmental regulations in the region go up by 1%, the level of green technology innovation in the region will rise by 0.099%. If environmental regulations in the surrounding areas increase by 1%, the region’s level of green technology innovation will go up by 0.124%. The overall effect will go up by 0.223%. The reason is that: (i) Environmental regulation is a mandatory binding force for enterprises to improve the ecological environment and meet the current needs of high-quality development. Hence, enterprises usually purchase a large amount of environmental protection equipment to comply with the new regulations to operate normally. Therefore, environmental protection costs rise and increase the total innovation capital. However, environmental regulation incentivizes green technology innovation and compensates for the previous. This verifies the validity of Porter’s hypothesis on green technology innovation and coincides with Hong Peng’s [65] view on the direct effect of environmental regulation on green innovation. (ii) Good environmental regulations will lead to good information, which will help businesses that focus on clean production to invest more in innovation. As a result, the government has put a lot of pollution control money into key industries. Investing more in research and development (R&D) to meet environmental standards and improve technology levels will lead to better financial indicators in the long run. At the same time, industries like cement, that are tightly regulated and limited, will slowly leave the market or be upgraded, pushing businesses to invest more in green technology innovation. (iii) In the context of high-quality development, the central government gradually improves how it evaluates the environmental performance of local governments, and competition between local governments takes the form of “top-by-top competition” [66]. When an effective environmental regulation policy is introduced in one region, neighboring regions follow suit [67]. At the same time, due to the fast and easy dissemination of information and knowledge externalities, environmental regulation leads to technological innovations being adopted and imitated, creating a free-rider effect [68]. Both geographic and economic proximity can positively influence local innovation, with the difference that geographic proximity is achieved mainly through the transfer of polluting firms. In contrast, economic proximity is promoted through a synergistic development path [69]. It is worth noting that the neighboring effect of environmental regulation is greater than the local effect. Environmental regulation more effectively promotes the development of green innovation in neighboring areas.

### 5.5. Robustness Test

This paper replaces spatial weights for robustness testing and chooses the neighboring weight matrix and economic weight matrix to verify the stability of the above research findings. The results are shown in Table 7 and Table 8, respectively. Except for a slight discrepancy in the estimated coefficients, the main explanatory variables were significant and directionally consistent with the outcomes. This suggests that the study’s findings are more reliable.

## 6. Conclusions and Policy Implications

Environmental regulation’s importance in green technology innovation is growing in the context of high-quality development. This paper uses panel data from 30 Chinese provinces and cities accumulated from 2009 to 2020 to examine environmental regulation’s direct impact and a spatial spillover effect on green technology innovation. The results reveal that: (i) Environmental regulation significantly affects green technology innovation, manifested by the increased intensity of environmental regulation. It not only helps to improve the level of green technology innovation in the region but also has a significant spatial spillover effect on green technology innovation in the neighboring regions. (ii) The learning effect of environmental regulation in the surrounding areas is greater than the direct effect of environmental regulation in the region.

This paper proposes the following countermeasures based on the above findings.

(i)The government should bring reforms in performance evaluation standards and strengthen incentives to improve green technology innovation, t. For example, environmental protection issues should be incorporated into the major indexes used to evaluate officials instead of relying solely on GDP. This would provide a robust, binding, and deterrent effect on regional officials. At the same time, by enhancing the importance given to evaluations of regional green technology R&D and by discouraging the polluting practices of businesses.(ii)The policymakers should consider the spatial spillover effect of environmental regulation. Regions need to work together to improve environmental governance. They must work together to investigate the possibility of establishing a coordinated development mechanism for regional ecological regulation. It will safeguard against the “negative effects of competition” carried about by competing for regional innovation and scientific and technological resources. In addition, it will remove obstacles to inter-regional environmental regulation policies.(iii)Reasonable planning of financial investments in environmental regulation, adjusting environmental governance to go deeper, putting the “Green Hills and Clear Waters Are Gold and Silver Mountains” development concept into practice, and giving full play to the role of financial functions. Moreover, strengthening the supervision of environmental pollution management funds ensures that plans run smoothly.

### Limitations and Future Directions

The study enlists the following limitations and directions for future researchers. (1) The study may have external validity threats as the spillover effect of environmental regulation intensity on innovation varies across regions. (2) The study lacks in analyzing much for non-coercive environmental regulations due to data concerns, and it can be a starting point for future studies. (3) Future studies can start by testing the heterogeneity of environmental regulation intensity in different regions using various proxy variables and making a deeper interpretation of the environmental regulation analysis.

## Figures and Tables

**Figure 1 ijerph-20-01069-f001:**
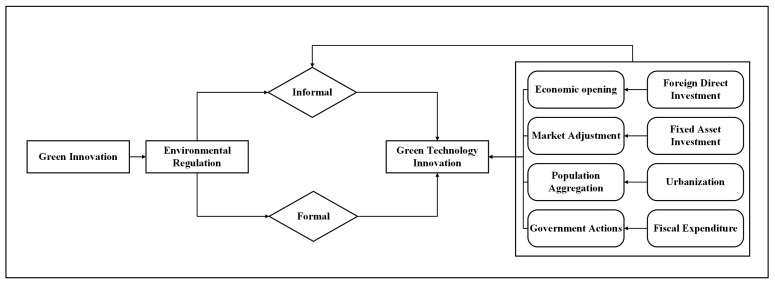
Variable association diagram.

**Figure 2 ijerph-20-01069-f002:**
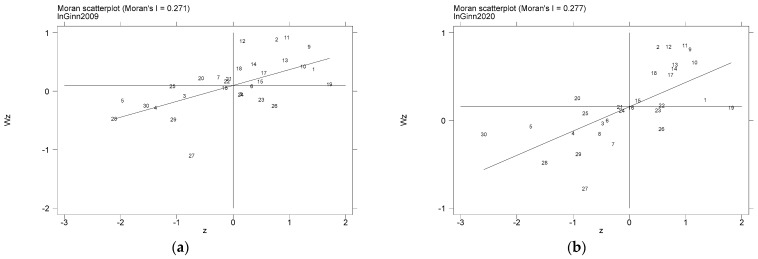
Green technology innovation scatter plot distribution: (**a**) 2009, (**b**) 2020.

**Table 1 ijerph-20-01069-t001:** Variable description.

Variable	Variable Description	Source
Ginno	Green Technology Innovation	China Statistical Yearbook
Ier	Environmental Regulation	China Statistical Yearbook
Fdi	Foreign Direct Investment	China Statistical Yearbook
Invest	Fixed Asset Investment	China Statistical Yearbook
Urb	Urbanization	China Statistical Yearbook
Fiscal	Fiscal Expenditure	China Statistical Yearbook

**Table 2 ijerph-20-01069-t002:** Descriptive statistics.

Variable	Observed Value	Average Value	Standard Deviation	Minimum Value	Maximum Value
Ginno	360	56.227	85.794	0.236	671.758
Ier	360	0.323	0.295	0.009	2.771
Fdi	360	0.02	0.016	0.001	0.082
Invest	360	0.773	0.248	0.21	1.48
Urb	360	0.577	0.128	0.299	0.896
Fiscal	360	0.245	0.102	0.096	0.643

**Table 3 ijerph-20-01069-t003:** Spatial correlation test.

Year	Moran’s *I*	Z-Value	*p*-Value
2009	0.271	3.192	0.001
2010	0.249	2.960	0.002
2011	0.287	3.382	0.000
2012	0.253	3.046	0.001
2013	0.269	3.222	0.001
2014	0.276	3.259	0.001
2015	0.275	3.251	0.001
2016	0.299	3.472	0.000
2017	0.272	3.208	0.001
2018	0.259	3.070	0.001
2019	0.252	3.008	0.001
2020	0.277	3.281	0.001

**Table 4 ijerph-20-01069-t004:** Results of spatial econometric model estimation.

Variable/Parameter	SDM	SAR	SEM
lnIer	0.097 ***	0.092 ***	0.096 ***
	(0.03)	(0.03)	(0.04)
lnFdi	−0.047	−0.099 **	−0.121 ***
	(0.04)	(0.04)	(0.04)
lnInvest	0.109	0.027	0.002
	(0.11)	(0.10)	(0.11)
lnUrb	1.052 *	3.166 ***	4.458 ***
	(0.55)	(0.41)	(0.33)
lnFiscal	−0.329	0.209	0.451 **
	(0.25)	(0.20)	(0.22)
Variance sigma2_e	0.107 ***	0.120 ***	0.129 ***
	(0.01)	(0.01)	(0.01)
N	360	360	360

Note: Standard errors are in parentheses, *, ** and *** indicate significance at the 10%, 5%, and 1% levels, respectively.

**Table 5 ijerph-20-01069-t005:** Results of spatial econometric model estimation.

Test Model	Statistical Value	*p*-Value	Sig
SAR	48.32	0.0000	***
SEM	67.11	0.0000	***

*** indicate significance at the 1% level.

**Table 6 ijerph-20-01069-t006:** Results of spatial panel model test.

Test Model	Statistical Value	*p*-Value	Sig
SAR	48.32	0.0000	***
SEM	67.11	0.0000	***

*** indicate significance at the 1% level.

**Table 7 ijerph-20-01069-t007:** Adjacency weight matrix.

Effect Category	Coefficient	Standard Error	T-Value	*p*-Value	95% Confidence Interval	
Direct effect	0.086 ***	0.037	2.350	0.019	0.014	0.158
Indirect effect	0.100 *	0.056	1.790	0.073	−0.009	0.210
Total effect	0.187 ***	0.055	3.370	0.001	0.078	0.295

*, *** indicate significance at the 10%, 1% levels, respectively.

**Table 8 ijerph-20-01069-t008:** Robustness results (economic weight matrix).

Effect Category	Coefficient	Standard Error	T-Value	*p*-Value	95% Confidence Interval	
Direct effect	0.099 ***	0.035	2.800	0.005	0.030	0.169
Indirect effect	0.124 **	0.062	2.000	0.046	0.002	0.245
Total effect	0.223 ***	0.065	3.450	0.001	0.096	0.350

** and *** indicate significance at the 5% and 1% levels, respectively.

## Data Availability

Data will be available on request to corresponding author.

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
