# Peer review of "The Empirical Analysis of Environmental Regulation’s Spatial Spillover Effects on Green Technology Innovation in China"

_ijerph, 2023, doi:10.3390/ijerph20021069_

Round 1
Reviewer 1 Report
The overall article is fine; however, I suggest authors address the following two minor issues.
1. For continuity, I suggest the author check the abstract; I feel they should improve or rewrite it.
2. literature is too long; I suggest making it a little short.
Author Response
We are thankful for the anonymous reviewer and editor comment. The manuscript is revised according to the reviewer’s comments. The point by point response is highlighted in the response and highlighted in the paper.
Comments and Suggestions for Authors
The overall article is fine; however, I suggest authors address the following two minor issues.
- For continuity, I suggest the author check the abstract; I feel they should improve or rewrite it.
Response: Thank you for the comment. The revised abstract as per the guidelines of reviewer.
- literature is too long; I suggest making it a little short.
Response: Thank you for the comment. Authors make it short and comprehensive as per advice. Please see literature section.
Reviewer 2 Report
This MS explore environmental regulation's direct and spatial spillover effects on green technology innovation in China. The authors have done a lot of work. However, several problems need to be revised. The following are my questions/comments.
General comments
1. There have been studies exploring the spatial spillover effects of environmental regulations on green technology innovation from the provincial level in China. What are the innovation points and knowledge gaps of your study compared to them?
Related articles:
Guo Q, Zhou M, Liu N, et al. Spatial effects of environmental regulation and green credits on green technology innovation under low-carbon economy background conditions[J]. International Journal of Environmental Research and Public Health, 2019, 16(17): 3027.
Dong Z, He Y, Wang H, et al. Is there a ripple effect in environmental regulation in China?–Evidence from the local-neighborhood green technology innovation perspective[J]. Ecological Indicators, 2020, 118: 106773.
Li X A. Environmental Regulation, Government Subsidies and Regional Green Technology Innovation[J]. Economic Survey, 2021, 38(3): 14−23(In Chinese)
2. Literature Review: The content of "The first view" is too long and needs to be expressed in a more concise manner. It is recommended that only classical and authoritative references be cited. In addition, the necessary logical lead sentence is missing. The summary of existing studies can be categorized according to the methodological and sample aspects.
3. Literature Review: The terms "formal environmental regulation" and "informal environmental regulation" have been proposed in the description of the shortcomings of the previous studies. However, they do not appear in the variable indicators in the later section. The description of the shortcomings of the previous study needs to be Improved.
4. The article lacks the necessary discussion section. An in-depth comparison of main results and previous studies should be conducted to elaborate on the most innovative findings and the knowledge gaps.
5. The findings support the " first view" in the Literature Review. Can you explain how the findings relate to other "views"? More attention needs to be paid to the comparison of "spatial spillover effects" with existing studies. Please give the necessary discussion.
Detailed comments
6. lines 22~23: "resource consumption, and industrial transformation" is a direct problem?
7. line 25: "The same year, China achieved the carbon peak", China has reached the carbon peak? Please give a reference?
8. line 67: "[8] used" lack of subject (name) before.
9. lines 219~221: There have been national and inter-provincial studies in China, see General Comment 1.
10. lines 289~291: The measure of green technology innovation (Ginn) needs to be given as a reference.
11. lines 293~295: Environmental regulation ( Ier ) metrics need to be referenced.
12. line 338 and line 430: Why is the content of Form 5 and Form 6 the same?
Author Response
We are thankful for the anonymous reviewer and editor comment. The manuscript is revised according to the reviewer’s comments. The point by point response is highlighted in the response and highlighted in the paper.
This MS explore environmental regulation's direct and spatial spillover effects on green technology innovation in China. The authors have done a lot of work. However, several problems need to be revised. The following are my questions/comments.
General comments
- There have been studies exploring the spatial spillover effects of environmental regulations on green technology innovation from the provincial level in China. What are the innovation points and knowledge gaps of your study compared to them?
Response: Thank you for the comment. The approach we applied in this study differs in the following way. The authors added the details in introduction and the last paragraph of literature review.
The previous studies tend to do only 0-1 matrix analysis. However, the actual spillover effect may not be limited to the neighboring areas. So the article uses three matrices to test the results (from adjusted weight and economic weight prospective). (2) The study also highlighted that provinces with higher levels of green technology innovation surrounded around provinces with higher levels of green technology. This phenomenon implies that green innovation is often not in the form of clusters. It is more dependent on the individual characteristics. (3) In the selection of control variables, this paper focuses more on the urban economic characteristics of the variables in contrast to the previous studies. Moreover, this study emphasizes the informal environmental regulation that conscious economic behavior can represent.
Related articles:
Guo Q, Zhou M, Liu N, et al. Spatial effects of environmental regulation and green credits on green technology innovation under low-carbon economy background conditions[J]. International Journal of Environmental Research and Public Health, 2019, 16(17): 3027.
Dong Z, He Y, Wang H, et al. Is there a ripple effect in environmental regulation in China? –Evidence from the local-neighborhood green technology innovation perspective[J]. Ecological Indicators, 2020, 118: 106773.
Li X A. Environmental Regulation, Government Subsidies and Regional Green Technology Innovation[J]. Economic Survey, 2021, 38(3): 14−23(In Chinese)
Response: The said articles are relevant and authors cited them into the article.
Please see line 131, 161 and 182.
- Literature Review: The content of "The first view" is too long and needs to be expressed in a more concise manner. It is recommended that only classical and authoritative references be cited. In addition, the necessary logical lead sentence is missing. The summary of existing studies can be categorized according to the methodological and sample aspects.
Response: The literature review was reduced for the number of literature cited. The theory of environmental cybernetics is also added. The marginal cost of control for firms under environmental regulation is introduced. The literature is categorized based on proof and heterogeneity analysis. The details are summarized in the last paragraph of said section.
- Literature Review: The terms "formal environmental regulation" and "informal environmental regulation" have been proposed in the description of the shortcomings of the previous studies. However, they do not appear in the variable indicators in the later section. The description of the shortcomings of the previous study needs to be Improved.
Response: We revised the section and explain the details about the informal regulations (line no.305 to 310).
The informal environmental regulation research is reflected in the choice of control variables. This idea is derived from Pargal's study. In Pargal's study, he argues against doing informal environmental regulation as a single variable. He focuses on the use of income and demographics as a proxy for informal environmental regulation. We inherited this idea in our study, but we chose some variables with more prominent economic implications.
“This paper refers Pargal's research [62], which measures the intensity of informal environmental regulation in each US state using a combination of variables. The research found that people's opinions about the environment improved with higher incomes, the transfer of government subsidies, exposure to the outside world, and higher population densities. For this reason, Foreign Direct Investment (FDI), Fiscal Regulation (Fiscal), Urban Regulation (Urb), and Environmental Investment (Invest) were selected as the intensity of informal regulation of the environment.”
- The article lacks the necessary discussion section. An in-depth comparison of main results and previous studies should be conducted to elaborate on the most innovative findings and the knowledge gaps.
Response: As per advice, the authors revised the results and discussion portion. As a result, we optimize for some of the empirical analyses. In particular, it illustrated our account of the centralized green innovation phenomenon. In addition, we have added new graphs and illustrations for some other innovations, including the choice of test methods and variables.
- The findings support the " first view" in the Literature Review. Can you explain how the findings relate to other "views"? More attention needs to be paid to the comparison of "spatial spillover effects" with existing studies. Please give the necessary discussion.
Response: as per advice, the authors explained/rewrite the results mainly by keeping the reviewer’s points in mind.
Detailed comments
- lines 22~23: "resource consumption, and industrial transformation" is a direct problem?
Response: Thank you for your valuable comment. The statement was confusing therefore; authors rewrite the statement for better understanding of readers. A: Change to “The Chinese economy has entered a new normal, with new and old problems such as overcapacity, low effective utilization of resources and challenging industrial upgrading.”
- line 25: "The same year, China achieved the carbon peak", China has reached the carbon peak? Please give a reference?
Response: The author rephrases the statement for better understanding and provided the reference.
“Carbon dioxide emissions in China are expected to reach their peak by 2030, and the government is working toward carbon neutrality by 2060 [1].”
[1] Wang, Y., Guo, C.H., Chen, X.J., Jia, L.Q., Guo, X.N., Chen, R.S., Zhang, M.S., Chen, Z.Y. and Wang, H.D., 2021. Carbon peak and carbon neutrality in China: Goals, implementation path and prospects. China Geology, 2021, 4(4), 720-746.
- line 67: "[8] used" lack of subject (name) before.
Response: Needful have been done as per advice. The order, reference and line numbers are changed due to extensive rewriting. It’s now on line number 95, “Mbanyele and Wang” [7] has been added.
- lines 219~221: There have been national and inter-provincial studies in China, see General Comment 1.
Response: the changes have been made as per advice.
- lines 289~291: The measure of green technology innovation (Ginn) needs to be given as a reference.
Response: Authors added the reference [60] as per advice.
Guo, J., The effects of environmental regulation on green technology innovation—Evidence of the porter effect in China. Finance & Trade Economics 2019, 3, 147-160.
- lines 293~295: Environmental regulation (Ier) metrics need to be referenced.
Response: As per suggestion we added the reference please see line no 301.
A: For the calculation of environmental regulation intensity (Ier), there are generally several ideas. One of them uses the cost of operating pollution control facilities, or the per capita operating cost to measure. Or it takes the change in pollution emissions or the intensity of pollution emissions per unit of output. However, all these calculation methods run the risk of having too homogeneous indicators. Therefore, this study uses the share of pollution control investment in the increase of enterprises as a proxy variable. This allows us to measure the degree of individual compliance with formal environmental regulations [61].
- line 338 and line 430: Why is the content of Form 5 and Form 6 the same?
Response: Thank you for highlighting it. We corrected the mistake. (See Line no. 447)
Reviewer 3 Report
The authors have examined the spatial spillover effect of environmental regulations on green technology innovation in China by using Spatial Durbin Model. Topic of the paper is quite interesting, and has significant contribution to the exisitng literature. However, I have few suggestions to improve the quality of the paper.
1. In the introduction section, the authors should clearly describe the importance and objectives of the study. All important innovations of this study should be highlighted, and clearly describe how this study is different from the previous studies.
2. In the literature review, the authors have just focused on Porter's hypothesis to explain the association between environmental regulations and technological innovations. It is not enough, I suggest adding more important theories
3. One conceptual framework diagram about the association between variables should be added.
4. One table should be added at the end of the methodology section, in which, measurement proxies, full names of variables, and data sources of each variable should be mentioned.
5. The results of descriptive statistics should be moved to the results section.
6. Estimated results of this study should be explained more in detail and correlate with the findings of previous studies.
7. The last section should be "Conclusion and Policy Implications". Limitations of the study and suggestions for future research should also be added in this section.
Author Response
We are thankful for the anonymous reviewer and editor comment. The manuscript is revised according to the reviewer’s comments. The point by point response is highlighted in the response and highlighted in the paper.
Comments and Suggestions for Authors
- In the introduction section, the authors should clearly describe the importance and objectives of the study. All-important innovations of this study should be highlighted, and clearly describe how this study is different from the previous studies.
Response: Thank you for the comment. The authors explain the importance and research objective in the last paragraph of introduction and literature. (Line no. 58-77)
This study will analyze the impact of environmental regulation and green technology innovation spillover effect for 30 provinces and cities in china. The study will add to the empirical literature in the following ways. Regarding research intention, environmental regulations related to green technological innovation are divided into formal and informal categories. Furthermore, the impact of environmental regulation and green technological innovation varies by province. So, examining the spillover effect to test regions' learning behavior is logical and can help in future decision-making. In terms of research content, the impact of environmental regulation on green technology innovation is limited to the regional perspective. Therefore, a spatial econometric model is used to investigate the spatial role of environmental regulations on green technology innovation from a broader viewpoint. The spatial Durbin model generates unbiased estimates of coefficients regardless of whether the correct data-generation procedure is a spatial lag or spatial error model. Furthermore, it does not limit the size of possible spatial spillover effects in advance, which is a positive aspect. At the same time, there is a tendency to simplify the use of matrices. This leads to a certain degree of bias in the analysis of spillover effects. This paper compares the regression results under the three matrices to demonstrate however, the previous studies tend to do only 0-1 matrix analysis. The spillover effects of environmental regulations. This study incorporated spatial factors into the model. Furthermore, we consider economic and spatial weights in the neighborhood as they may support enhancing green technology innovation.
- In the literature review, the authors have just focused on Porter's hypothesis to explain the association between environmental regulations and technological innovations. It is not enough, I suggest adding more important theories
Response: Thank you for the comment. Authors revised the literature section as per reviewer suggestion. The literature review was reduced for the number of literature cited. The theory of environmental cybernetics is also added. The marginal cost of control for firms under environmental regulation is introduced. The literature is categorized based on proof and heterogeneity analysis.
- One conceptual framework diagram about the association between variables should be added.
Response: Thank you for the comment. Authors added the diagram showing the association between the variables in the data section (Figure 1).
- One table should be added at the end of the methodology section, in which, measurement proxies, full names of variables, and data sources of each variable should be mentioned.
Response: As per advice we added the table (Table 1) in material and method section describing the variable and data source.
Table 1. Variable Description
|
Variable |
Variable Description |
Source |
|
Ginno |
Green Technology Innovation |
China Statistical Yearbook |
|
Ier |
Environmental Regulation |
China Statistical Yearbook |
|
Fdi |
Foreign Direct Investment |
China Statistical Yearbook |
|
Invest |
Fixed Asset Investment |
China Statistical Yearbook |
|
Urb |
Urbanization |
China Statistical Yearbook |
|
Fiscal |
Fiscal Expenditure |
China Statistical Yearbook |
- The results of descriptive statistics should be moved to the results section.
Response: Thank you for the comment. As per advice table is moved to results section on page no 8.
- Estimated results of this study should be explained more in detail and correlate with the findings of previous studies.
Response: Thank you for the comment. A: We provide a more detailed explanation of some of the more under analyzed empirical results. In particular, in for the Moran index.
- The last section should be "Conclusion and Policy Implications". Limitations of the study and suggestions for future research should also be added in this section.
Response: Thank you for the comment. The authors added the limitation and suggestions in the said section.
The study enlists the following limitations and directions for future researchers. (1) The study may have external validity threats as the spillover effect of environmental regulation intensity on innovation varies across regions. (2) The study lacks in analyzing much for non-coercive environmental regulations due to data concerns, and it can be a starting point for future studies. (3) Future studies can start by testing the heterogeneity of environmental regulation intensity in different regions using various proxy variables and making a deeper interpretation of the environmental regulation analysis.
Round 2
Reviewer 3 Report
I have reviewed the revised version, I feel satisfied with the revision submitted by the authors. These are in line with my suggestions. I suggest to accept this paper.